# External injuries, trauma and avoidable deaths in Agincourt, South Africa: a retrospective observational and qualitative study

Idara J Edem,[1] Anna J Dare,[2] Peter Byass,[3,4] Lucia D'Ambruoso,[4,5] Kathleen Kahn,[4] Andy J M Leather,[6] Stephen Tollman,[4] John Whitaker,[6] Justine Davies[7]

This work has been previously presented at the following conferences: AANS 2018, New Orleans, USA, and CNIS Bethune Round Table 2018, Toronto, Canada.

For numbered affiliations see end of article.

**Correspondence to**
Dr Idara J Edem;
iedem039@uottawa.ca

## ABSTRACT

**Objective** Injury burden is highest in low-income and middle-income countries. To reduce avoidable deaths, it is necessary to identify health system deficiencies preventing timely, quality care. We developed criteria to use verbal autopsy (VA) data to identify avoidable deaths and associated health system deficiencies.

**Setting** Agincourt, a rural Bushbuckridge municipality, Mpumalanga Province, South Africa.

**Participants** Agincourt Health and Socio-Demographic Surveillance System and healthcare providers (HCPs) from local hospitals.

**Methods** A literature review to explore definitions of avoidable deaths after trauma and barriers to access to care using the 'three delays framework' (seeking, reaching and receiving care) was performed. Based on these definitions, this study developed criteria, applicable for use with VA data, for identifying avoidable death and which of the three delays contributed to avoidable deaths. These criteria were then applied retrospectively to the VA-defined category external injury deaths (EIDs—a subset of which are trauma deaths) from 2012 to 2015. The findings were validated by external expert review. Key informant interviews (KIIs) with HCPs were performed to further explore delays to care.

**Results** Using VA data, avoidable death was defined with a focus on survivability, using level of consciousness at the scene and ability to seek care as indicators. Of 260 EIDs (189 trauma deaths), there were 104 (40%) avoidable EIDs and 78 (30%) avoidable trauma deaths (41% of trauma deaths). Delay in receiving care was the largest contributor to avoidable EIDs (61%) and trauma deaths (59%), followed by delay in seeking care (24% and 23%) and in reaching care (15% and 18%). KIIs revealed context-specific factors contributing to the third delay, including difficult referral systems.

**Conclusions** A substantial proportion of EIDs and trauma deaths were avoidable, mainly occurring due to facility-based delays in care. Interventions, including strengthening referral networks, may substantially reduce trauma deaths.

## INTRODUCTION

Injuries account for an estimated 5.8 million deaths per year, 32% more than HIV, malaria and tuberculosis combined.[1] The burden of trauma deaths is largest in low-income and middle-income countries (LMICs), which are also the most ill-equipped to manage this burden.[2] Although there has been a global drive for injury prevention and we are in the 'decade of action for road safety', morbidity and mortality from injuries continue to rise.[3] Alongside injury prevention efforts, optimisation of medical care provided after injury occurrence is important. In many LMICs, there are deficiencies in emergency medical services (EMS) and definitive hospital care for the injured. Where present, these are hampered by limited geographical coverage, resources and trained staff.[4]

### Strengths and limitations of this study

► As far as we are aware, this is the first time that verbal autopsy (VA) data have been used to ascertain avoidable mortality and access to care after trauma.
► As well, it is the first time that the three delays framework for access to emergency medical care has been systematically applied to trauma and avoidable mortality, allowing for the quantification of the delays to trauma care.
► Although this study was performed in a single Health and Socio-Demographic Surveillance System, the findings are novel, pertinent to the sustainable development goals, and the methodology can be more widely applied to other VA data worldwide.
► Key informant interviews of healthcare workers added useful information to the VA data, but this information is likely to be skewed towards healthcare facilities.
► The VA methodology involves interviews conducted by well-trained but non-medical personnel, and does not capture detailed vitals at the scene or the nature of prehospital and in-hospital emergency care received, which can result in missing important clinical aspects of the trauma and the care sought.

In order to improve a health system's ability to manage a large burden of injuries, gaps in care provision must be identified to guide resource allocation. To assist with this, the concept of completely or partially avoidable deaths—given the provision of effective healthcare—can be used[5] in association with exploration of the factors which led to an avoidable death occurring.

The three delays framework (delays in seeking, reaching and receiving care) for accessing safe, affordable and timely medical care is regarded as a classic conceptualisation of delayed care in emergency situations, reflected in its application in obstetric emergencies,[6] sepsis management,[7] perinatal care[8] and hip fractures.[9] To the best of our knowledge, the three delays model has not previously been analytically applied to trauma, but given the similarities between trauma and other time-critical conditions in which it has been used,[6] it is an appropriate model for investigating gaps in the health system contributing to avoidable trauma deaths.

To explore avoidable trauma deaths, a complete understanding of deaths and the events surrounding these deaths is needed. A lack of Civil Registration and Vital Statistics systems in many LMICs,[10] coupled with limited physician-assigned cause of death documentation, means that verbal autopsy (VA) has become a practical tool for ascertaining cause of death, including for injuries.[11] VAs have not previously been used to examine avoidable deaths in trauma.

Our aims were (1) to develop and validate criteria applicable to VA data to estimate the number of avoidable deaths in trauma; (2) in cases where deaths were avoidable, to develop criteria for assessing where in a three delays framework delays leading to deaths occurred, and estimate the number of cases where such delays were experienced; (3) perform qualitative interviews with healthcare workers to further explore delays to care; and (4) to combine information from the literature, VA and qualitative analysis to construct a three delays framework applicable to avoidable mortality in trauma.

## METHODS

### Literature review

To determine definitions and criteria for avoidable mortality and factors contributing to the three delays in trauma, PubMed, Ovid and EMBASE were searched using the terms 'trauma' AND 'avoidable death' OR 'preventable death' OR 'mortality' OR 'delays', and limited to papers published in English, from 1990 onwards, in high-income countries and LMICs. The titles and abstracts were reviewed to select papers addressing avoidable death in trauma and delays to care.

### Setting

The study was performed in Agincourt, located in the rural Bushbuckridge municipality of Mpumalanga Province, South Africa. A Health and Socio-Demographic Surveillance System (HDSS) has regularly collected household data on health and vital events (births, deaths, migrations) for over 115 000 people in a geographically defined area since 1992. All deaths identified during surveys are followed up and a VA is conducted within 1 year of the death.

### Verbal autopsy

VA involves interview of the lay primary caregiver of the deceased by a trained fieldworker about signs, symptoms and circumstances surrounding death. The interviews are completed within 12 months of the death, using a validated questionnaire, with good previously demonstrated recall.[12] Currently, a computer algorithm, InterVA-4, is used to determine cause of death, using standardised probabilistic models.[13]

VA data used in this study were collected using the WHO 2012 Short-Form.[14] In addition to standard medical questions to allow cause of death to be ascertained as well as information on treatment and healthcare utilization, the VA data contains a *trauma module*—41 questions to further query circumstances surrounding trauma deaths—and 10 'circumstances of mortality' (COM) questions that address the 'household, community and health systems determinants of health'[15] influencing mortality. In the trauma module, for each category of death, there are further specific questions to pinpoint the exact details of the deaths. COM questions are grouped under four themes—recognition of severity, mobilising assets to seek care, access to care and quality of care. Additionally, VAs contain a free-text summary of the circumstances surrounding deaths, obtained from the respondent, which allows information not captured in the survey questions to be ascertained.

Data from all VAs between 2012 and 2015 categorised as 'external injury deaths' (EIDs) (EIDs correspond to external causes of death in chapter 12 of the WHO 2012 VA instrument (WHO, 2012)) by InterVA-4 were extracted.[13] InterVA-4 classifies the following as EIDs: poisonings, drownings, deaths from natural disasters and 'traumas'—which, as a subclass of EIDs, include road traffic accidents, assaults, fires and falls. This broad VA category of external injury deaths, of which trauma is a subset, is referred to in the text as EIDs. Additionally, free-text searches of the open free-text summary portion of the VAs were performed to identify trauma deaths that may have been incorrectly categorised. Search terms included injury, trauma, fall, accident, traffic and fire.

### Defining avoidable death and developing a three delays framework

Findings from the literature review informed which variables contained in the VA questions and factors extractable from the free text had utility for defining avoidable deaths and in the creation of a three delays framework. In particular, each narrative in the free text was reviewed for details pertaining to location of external injury or death, acuity or chronicity of external injury, signs and symptoms after external injury, ability to seek care, factors

facilitating or inhibiting access to care, the provision of medical care, results of care delivered, and understanding of the effects of care on the patient's outcome.

VA reports on deaths from 2015 were first used to categorise deaths by avoidability and then delay. The categorisations of avoidability and three delays in a subsample of 20 cases were then validated by external review with a researcher and trauma care provider (JW) familiar with the geographical context. Discrepancies were discussed until consensus was achieved in order to improve classification of subsequent data. Once agreement was reached on categorising deaths by avoidability and then using the three delays framework, avoidable deaths and delays were discerned for all EIDs from 2012 to 2014. Where there were multiple delays contributing to the avoidable mortality, that which was found to be the most immediate factor was considered the primary contributing delay.

### Qualitative analysis

Purposive sampling was used to identify emergency care providers from local hospitals to participate in key informant interviews (KIIs), and snowball sampling was used until responses reached saturation and no new ideas were revealed.[16] The interviews were face-to-face, semistructured and open-ended. They were completed in English, recorded on tape, transcribed and transferred for analysis into NVivo V.11.4 (QSR International, 2017). Transcripts were analysed using framework methods[17] within a thematic analysis. This allowed for unanticipated meanings to emerge from the data and analysis, as well as for the data to be structured based on the overall aims and objectives of the study, thus permitting both inductive and deductive data analyses.[18] This was an iterative process, to ensure that the content of the narrative data was as fully represented as possible. Identified factors were then compared with those garnered from the literature review and the VA analysis.

### Final three delays framework

The findings from the literature review, VA and qualitative analyses were combined to construct a holistic three delays framework for assessing avoidable deaths in trauma.

### Statistical considerations

This is an exploratory study and a power calculation was not performed. Data are described. Analysis was performed using SPSS Statistics V.24 software, NVivo V.11.4 and Microsoft Excel.

### Patient and public involvement statement

Patients were not directly involved in this exploratory study and no new patient data were collected. However, although no patients were involved in this study, Agincourt employs community engagement officers to ensure that community members are aware of research being done in the area—even if community dwellers are not involved—and that results are fed back to the community.

## RESULTS

### Literature review

Fifty-seven papers which explored the conceptualisations of avoidable death in trauma and delays in access to care were identified, as were nine additional papers from review of references sections. Of these, 43 papers were included in the literature review. Despite differences in the definition of avoidable death in trauma, common themes included external injury survivability, deviations from delivery of optimal care and implication of errors in care delivery on the death of the patient. These themes are also reflected in the WHO guideline for the definition of avoidable death in trauma—note that this guideline definition also includes calculations of the probability of survival and injury severity scores.[2] Definitions of survivability which do not use injury severity scores were based on vital signs at the scene[19]; being awake at the scene[19–21]; being able to seek care after external injury; or having reversible, stable[22] or non-severe external injuries involving a single system.[19–21 23] Most studies used a combination of these definitions (see online supplementary appendix 1 for details on the approaches to determine the criteria for survivability).

Relevant components of the three delays model identified in the literature were as follows: deciding to seek care (socioeconomic and cultural factors, perceived accessibility and perceived quality of care; eg, previous experience with the healthcare system, perceptions of severity, transportation, costs, EMS transport protocols); identifying and reaching care (actual accessibility factors; eg, EMS accessibility and timeliness of response, prehospital care); receiving adequate and appropriate treatment (actual quality of care factors; eg, poorly staffed facilities and inadequate management) (online supplementary appendices 1 and 2); see online supplementary appendix 3 for the full list of articles included in the literature review.

### VA data

Figure 1 shows the number of VAs included in the study. Between 2012 and 2015, there were 3001 deaths; of these, 260 deaths were categorised as EIDs (8.7% of the total), of which 189 were due to trauma (table 1). The subcategory distribution for both EIDs and trauma deaths is shown in online supplementary appendix 4. Traffic and assault injuries were the most common causes of trauma deaths.

### Development of avoidable death and three delays criteria for use with VA

In 2015, out of 482 total deaths, there were 48 EID cases which were used to develop the criteria for determining avoidability. Descriptive information on survivability was the only criteria available. Factors indicating survivability were being alive at the scene, being able to seek help or non-severe, single-system injuries (table 2). Twenty (42%) EIDs were determined to be avoidable (table 1). The external reviewer agreed on classification of avoidability of death in 90% of cases.

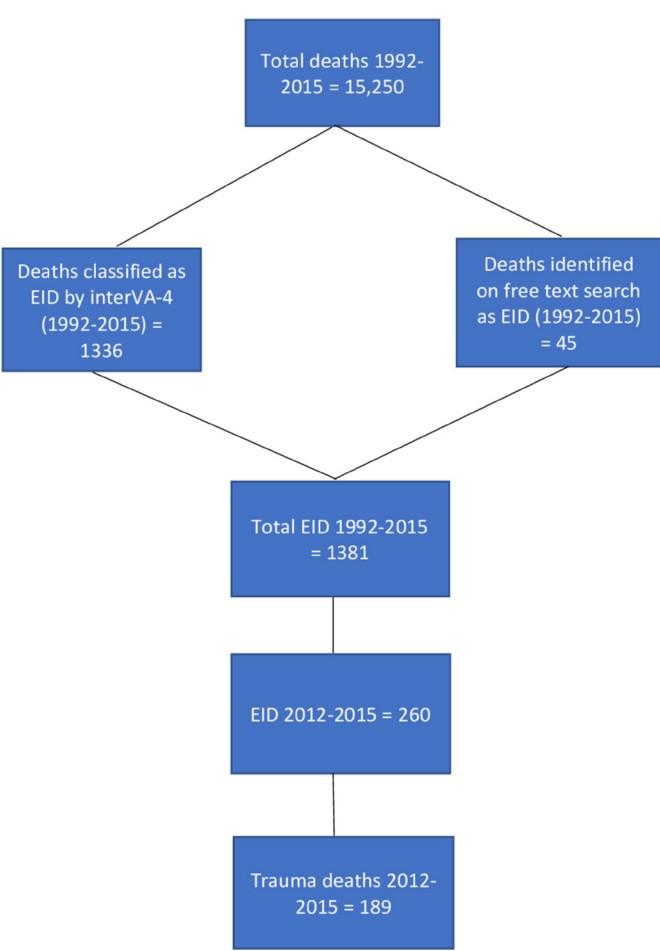

**Figure 1** Deaths considered in this study (EIDs and trauma), relative to total deaths in the Health and Socio-Demographic Surveillance System. EID, external injury death.

From these 2015 VA data, there were several extractable factors relevant to each of the three delays (table 2). There was 80% agreement with the external expert for classification of delays contributing to death. Further analysis and discussion of VA free-text data were performed together, until consensus was reached and a three delays framework using data relevant to VA was constructed. Data relevant only to the COM module are contained in online supplementary appendix 5.

### Avoidable mortality and three delays framework for EIDs and trauma, 2012–2015

Applying the definition of avoidable deaths to the 2012–2015 VA data revealed 104 avoidable EIDs (40%) and 78 avoidable trauma deaths (41%) (table 1, online supplementary appendix 4).

Based on the three delays framework, 58% of people with avoidable EIDs experienced one delay, 38% two delays and 4% three delays. For avoidable trauma deaths, 57% experienced one delay, 38% two delays and 5% three delays. For all avoidable deaths, perceived external injury severity, deficiencies in prehospital care, and inadequate and/or delayed diagnosis and/or treatments were the primary contributing factors in delays 1, 2 and 3, respectively (table 3). Delay 3 was the most commonly occurring contributing delay to avoidable deaths, in 61% and 59% of EIDs and trauma deaths, respectively.

When considering just the COM indicators (online supplementary appendix 5), these captured minimal data about the quality of care. Also, there were no cases where respondents reported having 'greater than 2 hours' travel to care' or 'doubts about the need for care'. However, a mobile phone was not used to call for help in 26% of

**Table 1** Demographic characteristics of the EID and trauma (a subset of EID) groups, for all, avoidable and non-avoidable deaths from 2012 to 2015

| Category | EIDs | Avoidable EID | Non-avoidable EID | Trauma deaths | Avoidable trauma deaths | Non-avoidable trauma deaths |
|---|---|---|---|---|---|---|
| Total | 260 | 104 | 156 | 189 | 78 | 111 |
| Age group | | | | | | |
| Older (>65 years) | 19 (7.3%) | 12 (11%) | 7 (4.5%) | 12 (6%) | 9 (11%) | 3 (2.7%) |
| Mid-age (50–64 years) | 27 (10%) | 10 (9.6%) | 17 (11%) | 20 (11%) | 8 (10%) | 12 (11%) |
| Adult (15–49 years) | 188 (72%) | 74 (71%) | 114 (73%) | 143 (76%) | 57 (73%) | 86 (78%) |
| Child (5–14 years) | 14 (5.4%) | 4 (3.8%) | 10 (6.4%) | 7 (3.7%) | 1 (1.3%) | 6 (5.4%) |
| Under 5 (1–4 years) | 10 (3.8%) | 3 (2.9%) | 7 (4.5%) | 5 (2.6%) | 2 (2.6%) | 3 (2.7%) |
| Infant (1–11 months) | 1 (0.4%) | 0 | 1 (0.6%) | 1 (0.5%) | 0 | 1 (0.9%) |
| Neonate (<28 days) | 1 (0.4%) | 1 (1.0%) | 0 | 1 (0.5%) | 1 (1.3%) | 0 |
| Sex | | | | | | |
| Male | 198 (76%) | 75 (72%) | 123 (79%) | 147 (78%) | 59 (76%) | 88 (79%) |
| Female | 62 (24%) | 29 (28%) | 33 (21%) | 42 (22%) | 19 (24%) | 23 (21%) |

EID, external injury death.

**Table 2** Details on the data sections of VAs used to define injury survivability and determine delays to care

| Description | VA sections containing relevant data | Variables extracted |
|---|---|---|
| Injury survivability | Demographics | Age. |
| | Acuity of death | Acute versus chronic injury/death. |
| | Medical history | List of medical conditions. |
| | Signs and symptoms | System-based list of signs and symptoms. |
| | External injury classification | Category of injury. |
| | Trauma module | Details of injury classification and mechanism of injury. |
| | Free-text disease descriptions | Vitals at the scene. Injured body system. Signs and symptoms at the scene and at hospital presentation. Injury severity. Actions taken immediately after injury. Ability to seek care. |
| First delay | Circumstances of mortality questions | Doubts about the need for care. Use of traditional medicine. Did not use a mobile phone to call for help. |
| | Free-text disease descriptions | Healthcare literacy and perceived severity of the injury. Perceived costs of seeking care—transport, medical costs. Perceived aetiology—trauma unwitnessed by caregivers or illness attributed to other known conditions. Perceived quality of care. Previous experiences with medical care. |
| | Trauma module | Traffic accident location, that is, rural versus urban. Death at site or time of injury. |
| Second delay | Circumstances of mortality questions | Did not use motorised transport to get to care. >2 hours travel to care. Prohibitive costs. Did not travel to a hospital or health facility. |
| | Free-text disease descriptions | EMS accessibility. EMS timeliness of response. EMS versus personal transport. Prehospital care. |
| Third delay | Circumstances of mortality questions | Problems with admission. Problems with treatment (medical treatment, procedures, interpersonal attitudes, respect, dignity). Problems with tests and medications. |
| | Free-text disease descriptions | Inadequate and/or delayed diagnosis and/or treatment. Delayed interhospital transfers. Poorly staffed and/or equipped facilities. |
| | Treatment and healthcare utilisation | Medical and/or surgical treatment. Traditional medicine treatment. If medical and traditional medicine treatment sought, which was sought first? Medical aid or employer support for healthcare. Medical actions. If no medical actions taken, reasons for not taking medical action. |

EMS, emergency medical services; VA, verbal autopsy.

EID cases, and 28% did not travel to a hospital for care. For avoidable trauma deaths, 27% did not use a phone to call for help, and 27% did not travel to a hospital for care. Also, issues with the prohibitive costs of seeking care were present in 20% of the avoidable EIDs and 21% of the avoidable trauma deaths.

**KIIs and qualitative analysis**

Seven KIIs were needed to reach saturation; these were done with two clinical associates (3-year training for a Bachelor of Clinical Medical Practice), four medical officers (4-year medical degree, 2-year internship, 1-year community service) and one casualty (emergency room)

**Table 3** Primary contributing factors in each delay category for avoidable EID and trauma deaths, 2012–2015*

| Delay | n (%) | Primary contributing delay factors | n (%) |
|---|---|---|---|
| Avoidable EIDs (n=109) | | | |
| 1—Seeking care | 25 (24) | Perceived severity | 35 (74) |
| | | Perceived aetiology | 6 (12.7) |
| 2—Reaching care | 16 (15) | Prehospital care | 28 (61) |
| | | EMS timeliness and response | 17 (37) |
| 3—Receiving care | 63 (61) | Inadequate and/or delayed diagnosis and/or treatment | 82 (78) |
| | | Delayed interhospital transfer | 11 (10) |
| Avoidable trauma deaths (n=81) | | | |
| 1—Seeking care | 18 (23) | Perceived severity | 23 (77) |
| | | Delayed discovery | 3 (9.6) |
| 2—Reaching care | 14 (18) | Prehospital care | 25 (63) |
| | | EMS timeliness and response | 13 (34) |
| 3—Receiving care | 46 (59) | Inadequate and/or delayed diagnosis and/or treatment | 59 (77) |
| | | Delayed interhospital transfer | 9 (11) |

*Note that one individual may experience more than one delay.
EID, external injury death; EMS, emergency medical services.

manager working in the three hospitals in Bushbuckridge: Matikwane, Tintswalo and Mapulaneng. Important themes included healthcare literacy, EMS versus personal transport, and mistrust of clinics (delay 1); EMS accessibility and timelines, and prehospital care (delay 2); and poorly staffed facilities, inadequately trained staff, lack of equipment and supplies, difficult referral systems, and staff morale (delay 3) (see below and online supplementary appendix 6 for full details on these themes. Note that an abridged version of the KII data analysis is presented here, but more details are presented in online supplementary appendix 6.)

### Seeking care
#### Health or healthcare literacy
Key informants (KIs) felt that most patients were unaware of the proper channels for seeking care, especially when an injury was not acutely severe.

> So, there are those who say straight to the hospital and those who say no but you can never actually tell which one because they don't know themselves…it's just a matter of their specific attitude, and there are mistakes in both ways…—Medical officer

#### EMS versus personal transport
KIs recounted that patients are aware of delayed responses from EMS, and previous experience with EMS delays may affect the decision to seek care.

> From what we've heard, it's delayed response from the ambulances, the emergency services…it's a problem…and also for the patients, I think because this place is surrounded by villages…if they don't have

transport, their own transport to come here, it becomes a bit of a challenge.—Clinical associate

### Reaching care
#### EMS accessibility and timeliness
KIs felt that the largest barrier for patients reaching care was EMS accessibility. Given the rural area and the limited number of ambulances, several descriptions were provided of patients waiting for long periods to be taken to hospital.

> …the biggest problem I think they have is access to emergency care personnel, the paramedics, the ambulance. People may have trauma cases and when they call, the ambulance is already out, attending to trauma care somewhere else. It might take two to three hours for this ambulance to come and pick this patient. Now that is just too long for some cases. That might mean the difference between being normal again and being invalid or even dead.—Medical officer

#### Prehospital care
As well as the issues with EMS timeliness, the care provided by EMS providers was described as limited by the training received, as well as by lack of equipment to provide early resuscitation.

> It's not only death, we can also prevent the sequelae people come in with…they get brain damage, because the care was too late. The thing of the transport is very important, because the other guys, the EMS, if something happens, they are not well trained to

resuscitate those patients. I've seen people die here because of that.—Medical officer

## Receiving care
### Inadequately trained staff

A key recurring theme was the need for more training. Many KIs reported that they did not receive standardised training in trauma care (eg, advanced trauma life support (ATLS) courses), and those that did expressed the lack of continued learning.

There is no specific ATLS course given to us, actually in the province as a whole…In other hospitals, they send their doctors, especially their junior doctors, to other places, to do their ATLS training, but with us, there is not that. So, you have to be organized and find out when an ATLS course is going to be, find out for yourself and pay for yourself. Which is not convenient also for the hospital…you cannot go away for a week…let's say it's me and my colleague who decide to go. That means for that week, we will leave a hole in the hospital.—Medical officer

### Lack of equipment and supplies

KIs reported that given financial constraints, there is often a lack of important medical supplies, and when they are present they easily become non-functional without being repaired or replaced on time.

…the second problem is life support equipment. Today it is there, tomorrow, it is broken down, yes… there is a blood corner, where we keep emergency blood, maybe two or three units in a fridge.—Medical officer

### Difficult referral system

A major problem in the care of severely injured patients, who are most commonly neurotrauma patients, is the ability to transfer them to a tertiary care institution, which is the only location for a CT scan and access to a neurosurgeon. KIs recounted that some of the barriers include miscommunication about where to transfer patients to and limited ICU ambulances.

Ok, the first, the big barrier, is the referral system. I can have a patient here, head injury, low GCS [Glasgow Coma Scale], they bring the patient in respiratory distress. I'll try to stabilize the patient…tube the patient…I'll get the doctors at the higher level, they will accept the patient, but to transfer that patient, I need an ICU ambulance but in the province, there are only two ICU ambulances…maybe it can take three to four hours…Then take the patient from here to [the other hospital], that is six hours, then it is a problem…especially since we don't have good ventilation equipment.—Medical officer

Well, sometimes, it's difficult because our referral system is contradictory. Like last week, we were

surprised that our referrals should be going to [this] hospital. Meanwhile, we have been arranging them with [another hospital]…They said that memo had been written in March, but we didn't know about it.— Medical officer

## Composite three delays framework for avoidable deaths in trauma

The creation of the final three delays framework for avoidable death in trauma based on the analysis of information from three methodologies (literature review, VA analysis and KIIs) is shown in figure 2.

## DISCUSSION

From 2012 to 2015, a large proportion of VA-defined EIDs were avoidable. Most avoidable EIDs occurred in male adults between 15 and 49 years old and were a result of traffic injury or assault. A delay in receiving definitive care (the 'third delay') was the largest contributing factor in avoidable deaths, and occurred as a result of delays in diagnosis and treatment, inadequate referral systems for injured patients, and poorly staffed and equipped health facilities, particularly at the district hospital level. These findings demonstrate that VA data examining the nature of the death and circumstances around death can be used to determine avoidable deaths and explore the factors contributing to these. Qualitative information from local healthcare providers (HCPs) provided useful additional information.

Globally, there is an increasing prevalence of trauma morbidity and mortality, especially in people younger than 45 years old,[24] with road traffic injuries being the second leading cause of death in those aged 15–49 years old and the first in men.[25 26] In Mpumalanga, road traffic injuries were the second leading cause of years of life lost in 2012, after HIV/AIDS.[27] Trauma burden is worse in LMICs, given the limited scope of trauma systems and the challenges of service delivery in already strained health systems.[4] When injuries do occur, it is important to have trauma systems in place that ensure that the injured person receives care from the time of injury occurrence, to reduce avoidable mortality. The logical way to develop such systems in countries with nascent healthcare systems is to first identify gaps in current services. To our knowledge, this is the first time that data from VAs have been used to do this. Findings from this study should contribute to future local health system planning and create a path for future studies at other HDSS collecting VA data.

Avoidable mortality was identified using data available from VA with criteria informed by a review of the literature. In trauma, there is heterogeneity in methods for categorising avoidable deaths, but survivability—defined using injury severity scores and expert panels—is the most common methodology.[28] Although survivability scores were not available in this study, VA contained data which were used—with good agreement—to define survivability. It is reassuring that identified themes here

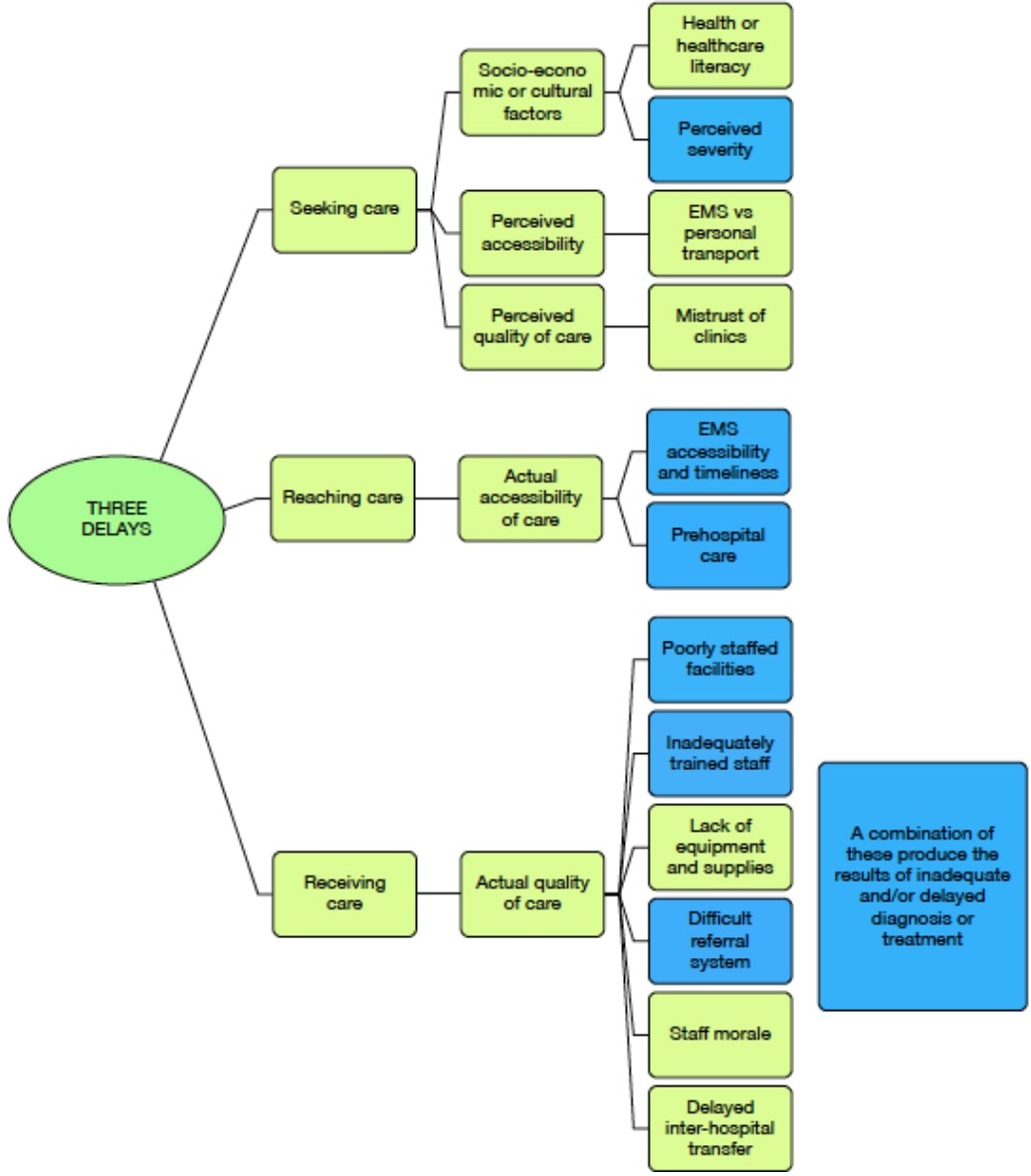

**Figure 2** Composite three delays framework for avoidable death in trauma. The most frequently cited factors from the verbal autopsy analysis and key informant interview analysis are shaded in blue. EMS, emergency medical services.

are also reflected in the WHO guideline for the definition of avoidable death in trauma. The results on the rates of avoidability are also similar to findings from other LMICs.[4 29–31] However, the avoidable death rate found in this study is greater than that found in high-income countries.[28 32–34] The findings that most avoidable deaths were in the group defined by VA as adults (between 15 and 49 years old) and in men are also consistent with previous studies on trauma deaths.[21 24 35]

In this study, delays in receiving care could not be ascertained from VA review alone, and KIIs provided useful additional data. HCPs revealed that factors including difficult referral systems, poorly staffed facilities, inadequately trained staff, and lack of equipment and supplies all contributed to delayed or inadequate diagnosis and/or treatment, and thus avoidable death. These factors have also been shown to be contributors to avoidable

trauma deaths in other studies,[36–39] and they represent possible intervention points. Potentially, the widespread introduction of trauma training, including ATLS and training provided to HCPs in LMICs by institutions like Primary Trauma Care Foundation, could lead to reduced trauma avoidable mortality, as has been seen in other studies.[40] A review of the interventions to improve trauma care quality in LMICs, their cost-effectiveness and possible application in this setting may be a worthwhile future study to pursue. Although the delays are presented linearly to enable understanding, they are actually interconnected, as delays in reaching and receiving care at one instance will influence the decision to seek care in future instances. Therefore, what is determined to be the most immediate factor contributing to death and thus the largest contributing delay may actually be the outcome of the effect of other delays in the past. As well, patients

cycle through the delays model as their care plans change and they require transfer to other facilities, such that these patients face new and compounding delays as their care provision progresses.

The result that a substantial proportion of mortality is avoidable seems at odds with the findings in a recent study showing that, in South Africa, only 5.2% of the population is estimated to be outside a 2-hour travel time to emergency care.[41] These results are reflected in the analysis of the COM data where there were no reports of being outside a 2-hour travel distance to care. Yet, on examination of other parameters, including the free text, delays in reaching care—especially *appropriate* care—did contribute to avoidable mortality. Indeed, avoidable mortality may be best reduced by introducing effective prehospital trauma care[36] and by ensuring that a bypass protocol is implemented,[42] so that patients with severe trauma injuries are transported immediately to secondary or tertiary hospitals without first being delayed in district facilities. Many severely injured patients in this study experienced a substantial delay when sent first to district hospitals before awaiting transfer to regional or tertiary hospitals. This was corroborated by information from VA narratives and KIIs with emergency care providers. The goal of a well-functioning trauma system should be that the 'right patient is treated at the right hospital at the right time'.[43] So, if patients are severely injured, they are taken directly to hospitals with appropriate specialists. This has the advantage of eliminating interhospital transfer delays. However, the success of a trauma system with targeted patient transfers also depends on an appropriately equipped and staffed prehospital care system, which is lacking in many LMICs, as shown here.

Neurotrauma was a significant cause of avoidable trauma deaths in this study, as seen both from findings of the KIIs and VA review, where 35% of the avoidable trauma deaths had neurotrauma. This finding is likely an underestimation, given the lack of detailed medical information. Other studies have also found that most of the avoidable deaths in trauma are due to neurotrauma,[19 44] usually related to poor airway management.[24 45] In Bushbuckridge, where access to a neurosurgeon occurs only in a tertiary hospital outside of the municipality, it is paramount to provide adequate prehospital care, including proper airway management, as well as reduce interhospital transfer delays.

This study had several limitations. VA data do not capture detailed vitals at the scene, exact prehospital delays, or the nature of prehospital and in-hospital emergency care received. Therefore, although non-severe, single-system injuries were used as criteria for avoidable deaths, many more deaths could have occurred due to missed major secondary injuries, information that may not be well captured in VA data. Furthermore, the COM indicators provided information for the delays, but they did not capture important information about delays in reaching appropriate or quality care. The use of these indicators in South Africa was analysed in the census rounds of 2012–2013, and the researchers found that problems with not calling for help or not going to a medical facility were more pronounced than problems with overall costs in the younger age groups and in those with acute conditions, as in most trauma cases.[15] In this study, KIIs added useful information to the VA data, but this information is likely to be skewed towards healthcare facilities. If other service providers or users were interviewed, further information about delays to access in care which is not contained in VA data may have been gleaned. An additional limitation is that the acute nature of trauma may also bias caregivers to report the care provided in hospital in more detail, as was seen in the free text, so that if not asked specifically about the prehospital delays these are not recollected or recorded. The VA methodology involves interviews conducted by well-trained but non-medical personnel, which can result in important clinical aspects of the injury and the care sought being missed. Since the VA interviews are conducted months after the death of the individual, recall bias becomes a major factor, affecting the reliability and validity of the information provided. Yet, although the VA is not the ideal tool for information on death and causes of death, it is a pragmatic method that provides useful information, and its use continues to widen with greater innovations for its application.[15]

## CONCLUSION

In this study, VA was shown to be a feasible method for defining avoidable deaths in trauma and ascertaining which of the three delays contribute to death. Between 2012 and 2015, a large percentage of VA-defined external injury (40%) or trauma (41%) deaths were avoidable. The third delay in the three delays model was found to be the largest contributor to avoidable deaths, and the qualitative study supported the findings from the VA. To combat the burden of avoidable trauma deaths in LMICs, there is a need for a functioning trauma system, with adequate health systems interventions in trauma prevention, access to medical care, EMS, and adequately staffed and supplied hospitals. 'When it comes to trauma care, where people live ought not to determine if they live'.[46]

**Author affiliations**

[1]Department of Surgery, Division of Neurosurgery, University of Ottawa, Ottawa, Ontario, Canada
[2]Department of Surgery, University of Toronto, Toronto, Ontario, Canada
[3]Umeå Centre for Global Health Research, Umea Universitet, Umeå, Sweden
[4]Medical Research Council/Wits University Rural Public Health and Health Transitions Research Unit, Faculty of Health Sciences, School of Public Health, University of the Witwatersrand, Johannesburg, South Africa
[5]Centre for Global Development and Institute of Applied Health Sciences, University of Aberdeen School of Medicine and Dentistry, Aberdeen, UK
[6]King's Centre for Global Health, King's Health Partners and King's College London, London, UK
[7]Centre for Applied Health Research, University of Birmingham, Birmingham, UK

**Acknowledgements** We thank Chodziwadziwa Kabudula (MRC/Wits Rural Public Health and Health Transitions Research Unit—School of Public Health, Faculty of Health Sciences, University of the Witwatersrand, Johannesburg/Acornhoek, South Africa) for his assistance with assembling the Agincourt HDSS data set for our use.

The research presented in this paper was in part funded by the Health Systems Research Initiative from the Department for International Development (DFID)/ Medical Research Council (MRC)/Wellcome Trust/Economic and Social Research Council (ESRC) (MR/P014844/1).

**Contributors** IJE helped to develop the idea, did the analysis, drafted and revised the paper. AJD helped to develop the idea and inputted into the paper drafts. PB inputted into the idea and the paper drafts. LD inputted into the idea, supported the analyses and inputted into the paper drafts. AJML, KK and ST inputted into the idea and the paper drafts. KK set up and is overall responsible for the system of verbal autopsies. ST set up and is overall responsible for the Agincourt HDSS. JW contributed to the analyses and inputted into the paper drafts. JD led the project, developed the idea, supported the analyses and inputted into the paper drafts. All authors approved the final paper for submission.

**Funding** The research presented in this paper was funded by the Health Systems Research Initiative from the Department for International Development (DFID)/ Medical Research Council (MRC)/Wellcome Trust/Economic and Social Research Council (ESRC) (MR/P014844/1). A travel scholarship to enable this project was provided by the King's College London, Centre for Global Health. Funding MR/ P014844/1 has previously been used to develop the Verbal Autopsy data used in this study. These funding sources were not involved in this study design; in the collection, analysis and interpretation of the data; in the writing of the manuscript; and in the decision to submit the paper for publication.

**Competing interests** None declared.

**Patient consent for publication** Not required.

**Ethics approval** Ethics approval from the University of the Witwatersrand Human Research Ethics Committee to use data from the VAs in the Agincourt HDSS in secondary analyses had been previously obtained (M960720 and M110138). All analysed VA data were anonymised and no individual deceased person or respondent was identifiable from the data used and presented. Ethics approval for the independent KIIs was obtained from the University of the Witwatersrand Human Research Ethics Committee (M170269) and King's College London Research Ethics Committee (LRU-16/17–4313). Participation in the qualitative study was voluntary and informed consent was obtained for the interview and its recording. All interviews were confidential and narratives were anonymised.

**Provenance and peer review** Not commissioned; externally peer reviewed.

**Data sharing statement** The data used for the VA analysis are available from the INDEPTH Data Repository platform at http://www.indepth-ishare.org/index.php/ home. The supplementary file contains further details on the qualitative interviews. Further data analysis is available by emailing iedem039@uottawa.ca

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
