## [Reviewer comments · BMJ Open]

ARTICLE DETAILS

TITLE (PROVISIONAL)	EXTERNAL INJURIES, TRAUMA AND AVOIDABLE DEATHS IN AGINCOURT, SOUTH AFRICA: A RETROSPECTIVE OBSERVATIONAL AND QUALITATIVE STUDY
AUTHORS	EDEM, IDARA; Dare, Anna; Byass, Peter; D'Ambruoso, L; Kahn, Kathleen; Leather, Andy; Tollman, Stephen; Whitaker, John; Davies, Justine

VERSION 1 - REVIEW

REVIEWER	Stacy Drake The University of Texas Health Science Center at Houston Cizik School of Nursing
REVIEW RETURNED	06-Dec-2018

GENERAL COMMENTS	Thank you for the opportunity to review this interesting manuscript. The topic is relevant and timely for all of global healthcare. Overall, the manuscript was complex and difficult to follow. The data presented did not align with the methods or discussion points. Given the medicolegal death investigation system and autopsy services function different than other countries, I suggest describing the system in greater detail within the "setting" section. For example, that the medicolegal death investigation system allows the primary care physician to sign a trauma related death certificate is very much an anomaly in the UK or US. I did appreciate the definition section. However, the authors used the words: avoidable, preventable and survivable throughout to mean the same thing, I believe. I did not know that the study was a qualitative design until reading page 6. Another words, the abstract is confusing and misleading. Methods – As I reviewed the manuscript it felt like the authors were reporting 4 separate studies: retrospective review of deaths; qualitative interviews; a literature review and a 2015 pilot study. It was difficult to follow. I would recommend a methods manuscript and report the pilot data within the development manuscript. A framework manuscript. Followed by another manuscript describing the results of the implementation of such a tool/framework. Some specific examples are below: Page 4: the authors report only two aims; however, within each aim are several additional aims. Aim 1 has at least 4. Developing the criteria based on literature; experts is one. Validating the criteria; which requires statistical analysis is two. Followed with implementing the criteria/tool to identify the "avoidable deaths". Lastly, followed with a project to determine the barriers to care.
---

	Aim 2 is not written clearly. I believe the authors were attempting to develop a framework???Which requires a far more extensive literature review then presented. The analysis plan only addressed the qualitative interviews; however, who/how saturation was determined and how interviews were recorded and coded was unclear. p. 7. I am unclear how the authors had 260 external injury deaths (not clearly defined; however, implies the body had "injury present on external") and then went to 189 as the inclusion number? Based upon what variables? Injury=trauma. Thus, the inclusion/exclusion criteria was not described for the quantitative method section. The results requires the reader to rapidly switch between the different designs and are not presented clearly. I have questions/concerns about how a primary care provider months after death are able to answer some of the questions posed on p. 8-10. There is an assumption that the PCP actually asked the patient these questions. I am also unclear of who/how the details of injury classification are obtained, if an actual autopsy is not conducted. Is the reader to assume the patient didn't seek care due to lack of vehicle or did they all have a vehicle readily available? Some of the information presented within discussion was not provided in the results. P. 15 Line 52 "Many severely injured patients...."No presented data supports this statement. The variables needed to calculate this "delay" are time of injury; time of admission; or time provider made contact with patient". None of these variable appear to be present. P. 16 line 56 EMS are prehospital care providers
--	---

REVIEWER	Dr Simon Hendel Anaesthesiologist and Trauma Consultant, Department of Anaesthesia and Perioperative Medicine, The Alfred Hospital, The Alfred Trauma Service, and the National Trauma Research Institute Melbourne, Australia
REVIEW RETURNED	08-Dec-2018

GENERAL COMMENTS	The reviewer provided a marked copy with additional comments. Please contact the publisher for full details.
--

REVIEWER	Robert Larsen Department of Clinical and Experimental Medicine, Division of Surgery, Orthopedics and Oncology The Faculty of Medicine and Health Sciences, Sweden
REVIEW RETURNED	29-Jan-2019

GENERAL COMMENTS	Dear authors, thank you for the opportunity to to read your paper in an interesting subject. Bellow are my comments. Introduction: To help clarify your aims, please make it a separate paragraph. Methods: Settings: I would be very interested in why you chose Agincourt? Your Affiliations is Toronto, , Umeå, and London. Literature review: I would like to have the complete search string as a supplemental file, and I think that you should publish "search-
--

	date", how many articles that were included in the first search-string, how many that were screened, and exclusion criteria. Verbal Autopsy: please be more consistent you refer to InterVA4 and INTERVA-4. Stick to a name and use it throughout so the reader doesn't need to go back and check if it's any difference. Last sentence: Why didn't you include "traffic" in search terms? You do report data for traffic in the manuscript. Results: Literature review: I would like to see the reference list for those 57+9 papers, could be a supplemental but still needs to be addressed as a datasource. Do you have any comments on the difference between the unavoidable and avoidable deaths (the numbers are practically the same in table 1)? Discussion: The third sentence. Largest contributor in avoidable death. This is true in the unavoidable deaths too. Do you have any comments or reflections on this? Ethical consent is not discussed at all. I think you should comment on why you don't need the ethical consent and if you think it is correct. As the patients need to be identified for VA to be productive. However, VA involves diseased patients. I would like to thank you again for providing a well-written, easy-to-read manuscript in an interesting area...
--	--

VERSION 1 – AUTHOR RESPONSE

RESPONSE TO REVIEWERS

REVIEWER 1

I did appreciate the definition section. However, the authors used the words: avoidable, preventable and survivable throughout to mean the same thing, I believe.

Thank you. We have changed this and to be consistent with our terminology throughout, we have used 'avoidable'.

I did not know that the study was a qualitative design until reading page 6. Another words, the abstract is confusing and misleading.

We have now made it clear in the title and the abstract that this study involved qualitative work.

Methods – As I reviewed the manuscript it felt like the authors were reporting 4 separate studies: retrospective review of deaths; qualitative interviews; a literature review and a 2015 pilot study. It was difficult to follow. I would recommend a methods manuscript and report the pilot data within the development manuscript. A framework manuscript. Followed by another manuscript describing the results of the implementation of such a tool/framework. Some specific examples are below:

Page 4: the authors report only two aims; however, within each aim are several additional aims. Aim 1 has at least 4. Developing the criteria based on literature; experts is one. Validating the criteria; which requires statistical analysis is two. Followed with implementing the criteria/tool to identify the "avoidable deaths". Lastly, followed with a project to determine the barriers to care. Aim 2 is not written clearly.

We have adjusted the introduction section with the aims of the study for further clarification. We agree that there are 4 components in the study, however, these are linked, with each building on the one before. This is a formative paper, although with an extensive analysis done from development of concept to results in just one centre. To split it into four separate papers would reduce the impact of the whole. We would therefore prefer the manuscript to stay as one, rather than four, as suggested. We have made changes to address the issues of readability and have made alterations to the text and abstract to improve flow.

The analysis plan only addressed the qualitative interviews; however, who/how saturation was determined and how interviews were recorded and coded was unclear.

This information has now been added to the manuscript (page 6).

p. 7. I am unclear how the authors had 260 external injury deaths (not clearly defined; however, implies the body had “injury present on external”) and then went to 189 as the inclusion number? Based upon what variables? Injury=trauma. Thus, the inclusion/exclusion criteria was not described for the quantitative method section.

This has now been further clarified in the manuscript. Essentially, external injuries include poisonings, drownings, deaths from natural disasters and traumas—which as a subclass of EIDs, include road traffic accidents, assaults, fires and falls. See page 5.

The results require the reader to rapidly switch between the different designs and are not presented clearly. I have questions/concerns about how a primary care provider months after death are able to answer some of the questions posed on p. 8-10. There is an assumption that the PCP actually asked the patient these questions. I am also unclear of who/how the details of injury classification are obtained, if an actual autopsy is not conducted. Is the reader to assume the patient didn't seek care due to lack of vehicle or did they all have a vehicle readily available?

The information presented in the verbal autopsy is garnered from lay-respondents who were the primary caregivers to the deceased before death. The questions asked relate to the causes of death, circumstances of mortality surrounding the event, as well as the journey to seek and receive care after the injury. There are various reasons why people do not seek care after an injury, as is described in the delays to care framework. Given that in many LMICs, there is no death registration, a pathological autopsy is not performed, and there are no trauma registries, VA provide one of the only means to discern detailed information about deaths due to trauma. In situations where verbal autopsies are available, they have been shown to be a validated, useful tool to aid in beginning to understand the factors involved in death and prevention of death, at individual and systems levels.

We have made it clear (page 5) that the primary caregiver interviewed by VA is a lay person, and have further adjusted the description of VA methods to make them more understandable to readers who are not familiar with the methodology.

P. 15 Line 52 “Many severely injured patients...”No presented data supports this statement. The variables needed to calculate this “delay” are time of injury; time of admission; or time provider made contact with patient”. None of these variables appear to be present.

Although these variables are not quantitatively collected in the area in which the study was performed, the narrative sections of the VA do capture information which can be used to ascertain whether or not there were such delays. For instance, caregivers describe severely injured patients with brain injuries who were sent to district hospitals but needed transfer to tertiary hospitals, where neurosurgeons were located. Unfortunately, some died while they were waiting to be transferred, as the process could take days. In addition KIs described such delays being common in their experience, with acute brain injury patients dying in the district hospitals, while waiting transfer to tertiary hospitals (see page 14). This has been addressed in the manuscript.

P. 16 line 56 EMS are prehospital care providers

This has been addressed in the manuscript.

REVIEWER 2

Page 5 line 41—Are these not also trauma deaths?

The definition and classification of deaths, according to InterVA4 and the classification used in this manuscript have been further clarified (see page 5). Essentially, deaths from natural disasters are classified as external injuries, but not necessarily as traumas, as various causes of death could arise after natural disasters that are not necessarily traumatic in nature. We have maintained the same classification as InterVA4 to aid with understanding and comparison to other VA data.

Page 6 line 19—Do you mean purposely?

This was used to mean that purposive sampling was used initially to identify key informant interviewees (see page 6).

Page 8 line 16—How can you be sure in a basic or non-existent pre-hospital system that the diagnosis of “non-severe single system injury” is accurate? i.e. presumably some death is occurring due to missed major secondary injuries?

We acknowledge that non-severe single-system injuries was one criteria used to assess avoidable mortality and this has been used in other studies of avoidable mortality. However, we do acknowledge in the discussion section that these patients could later develop severe secondary injuries, which a verbal autopsy may not capture, if the respondent was not made aware of these injuries (see page 16).

Page 11 line 48—While all of the acronyms have been defined, using non-standard acronyms throughout reduces the readability of the paper.

Thank you. We have written the acronyms for COM indicators out in full, to aid with readability.

Page 15 line 52—This is a key and very important point to make. It is an accepted part of improving trauma care to individuals that a trauma system is built to optimize delivery of patients to the right place. The Victorian Trauma System in Australia is one such model that is relevant (although, of course, there are many acceptable specialized trauma systems).

Thank you. We do agree that this is an essential point in efforts to improve trauma care in LMICs.

REVIEWER 3

Introduction: To help clarify your aims, please make it a separate paragraph.

This has been addressed in the manuscript (page 4).

Methods: Settings: I would be very interested in why you chose Agincourt? Your Affiliations is Toronto, Umeå, and London.

Five of the authors work and perform research in Agincourt, as is listed in the affiliations. Actually, two of the authors are founding members of the Agincourt HDSS and have this as their sole affiliation.

Literature review: I would like to have the complete search string as a supplemental file, and I think that you should publish "search-date", how many articles that were included in the first search-string, how many that were screened, and exclusion criteria.

This has been addressed in the manuscript and has been added as a supplementary file.

Verbal Autopsy: please be more consistent you refer to InterVA4 and INTERVA-4. Stick to a name and use it throughout so the reader doesn't need to go back and check if it's any difference.

Thank you. This has been addressed in the manuscript.

Last sentence: Why didn't you include "traffic" in search terms? You do report data for traffic in the manuscript.

Apologies for the omission, we have now corrected the manuscript (page 5). Traffic and accidents were included in the search terms. This search was also only performed to garner any cases that may have been incorrectly categorized. Most of the cases had already been correctly labeled within the external injury categories.

Results: Literature review: I would like to see the reference list for those 57+9 papers, could be a supplemental but still needs to be addressed as a data source.

This information is included in appendices 1 and 2, to reflect the papers that were garnered from the literature review to address criteria for injury survivability and delays to care. We believe this is a good way to present the information, so when the papers are cross-referenced in the manuscript, readers can automatically see which were related to each aspect of the survivability criteria and the delays to care. We have presented one reference list for both the main manuscript and the appendices, but would be happy to split them (one for each of the manuscript and appendices) if the editors would prefer.

Do you have any comments on the difference between the unavoidable and avoidable deaths (the numbers are practically the same in table 1)?

With apologies for any misunderstanding in what the reviewer is referring to, but for each of the categories, EID and trauma, the numbers of non-avoidable deaths were substantially greater than avoidable deaths (156 vs 104 for EID, and 111 vs 78 for trauma). Please do let us know if we have misinterpreted this comment. As for the categorizations of avoidable and unavoidable deaths, given the mechanism of injuries and the previously used definitions for avoidability of death, the unavoidable deaths mostly reflect those who had more severe injuries.

Discussion: The third sentence. Largest contributor in avoidable death. This is true in the unavoidable deaths too. Do you have any comments or reflections on this?

We only applied the three delays framework to individuals who we defined as having experienced an avoidable death. We did not analyze delays in those whose death was unavoidable. Given the reliance on injury severity to determine avoidability in this study, it may be likely that earlier access to skilled pre-hospital care could have improved survival. These would be classified as delays in seeking care and reaching care. However, to speculate thus would be beyond the scope of this study.

Ethical consent is not discussed at all. I think you should comment on why you don't need the ethical consent and if you think it is correct. As the patients need to be identified for VA to be productive. However, VA involves diseased patients.

We did include a paragraph on ethical approvals, but have expanded on this in the revision (page 7).

VERSION 2 – REVIEW

REVIEWER	Stacy Drake The University of Texas Health Science Center at Houston
REVIEW RETURNED	06-Mar-2019

GENERAL COMMENTS	Overall, I think this is an important topic and appreciate the authors pay attention to reviewers suggestions. I am still not clear on the authors use of terms that essentially mean the same, e.g. trauma from injury and vice versa. I have provided some suggestions below. Abstract: "...avoidable deaths after external injury (including trauma)..." deaths after external injury including trauma...this is redundant. External injury can only be caused from a traumatic mechanism... p. 4 line 45deaths in trauma and other external injures;/..." this is somewhat confusing trauma / injuries are one in the same. What are the other external injuries if not from trauma and/or vice versa? Should be deaths from trauma. Even within the lay community injury and trauma have overlapping definitions and are at times inseparable. It is up to the researchers to help the reader be clear about how they are being used. trau·ma Dictionary result for trauma /'troumə, 'trōmə/ noun 1. 1. a deeply distressing or disturbing experience. "a personal trauma like the death of a child" o 2. 2. MEDICINE physical injury. synonyms: injury, damage, hurt, wound, wounding, sore, bruise, cut, laceration, lesion, abrasion, contusion "the gallstone can be extracted without unnecessary trauma to the liver" in·ju·ry Dictionary result for injury /'inj(ə)rē/ noun 1. an instance of being injured. "she suffered an injury to her back" synonyms: wound, bruise, cut, gash, tear, rent, slash, gouge, scratch, graze, laceration, abrasion, contusion, lesion, sore; technicaltrauma "he was taken to hospital with minor injuries" o the fact of being injured; harm or damage. "all escaped without serious injury" synonyms: harm, hurt, wounding, damage, pain, suffering, impairment, affliction, disablement, incapacity, disability; disfigurement "they are reasonably safe from personal injury"
---

	o offense to. "the possible injury to the feelings of others" synonyms: offense, abuse; More p. 5 the inclusion/exclusion criteria is confusing "...classifies as EIDs, poisonings, drownings, deaths from natural disasters and traumas—which, as a subclass of EIDs.. Maybe clarify with a statement about that EIA includes the following...causes of death from trauma, drowning, poisoning, etc.. clarifying the inclusion/exclusion would help with organizing the results. I get hung up on trying to distinguishing EIA's from injury...Table 1. SUGGEST ORGANIZING IN THE FOLLOWING FORMAT Category EIDs (WHICH DEATHS ARE THESE...CAR CRASHES, FALLS, ETC...INJURY RELATED Avoidable EID [MOVED UP] Nonavoidable EID [MOVED UP] Trauma deaths (WHICH DEATHS ARE THESE DROWNINGS, POSIONINGS, NON INJURY RELATED?) Avoidable trauma deaths [MOVED] Nonavoidable trauma deaths [MOVED]
--	--

VERSION 2 – AUTHOR RESPONSE

RESPONSE TO REVIEWERS

Abstract: "...avoidable deaths after external injury (including trauma)..." deaths after external injury including trauma...this is redundant. External injury can only be caused from a traumatic mechanism...

External injury deaths, as used in the manuscript, reflect a category of death classification used in verbal autopsy data in the data collected from multiple communities worldwide and in the InterVA-4 analysis of this data. We have tried to maintain the use of this category, to allow for better understanding and clarification of deaths, but we have also made changes to the manuscript to clearly state when this VA-defined external injury category is being utilized.

p. 4 line 45deaths in trauma and other external injures;/..." this is somewhat confusing trauma / injuries are one in the same. What are the other external injuries if not from trauma and/or vice versa? Should be deaths from trauma.

Even within the lay community injury and trauma have overlapping definitions and are at times inseparable. It is up to the researchers to help the reader be clear about how they are being used.

Changes have been made in the manuscript to clearly differentiate the use of VA-defined external injury deaths and trauma. Please see the explanation above.

p. 5 the inclusion/exclusion criteria is confusing "...classifies as EIDs, poisonings, drownings, deaths from natural disasters and traumas—which, as a subclass of EIDs.. Maybe clarify with a statement about that EIA includes the following...causes of death from trauma, drowning, poisoning, etc..

clarifying the inclusion/exclusion would help with organizing the results. I get hung up on trying to distinguishing EIA's from injury...Table 1.

InterVA-4 classifies the following as EIDs: poisonings, drownings, deaths from natural disasters and "traumas". Trauma is a subset of EIDs and includes road traffic accidents, assaults, fires and falls. We have clarified in the text when using the VA-defined EIDs.

SUGGEST ORGANIZING IN THE FOLLOWING FORMAT

Category

EIDs (WHICH DEATHS ARE THESE...CAR CRASHES, FALLS, ETC...INJURY RELATED

Avoidable EID [MOVED UP]

Nonavoidable EID [MOVED UP]

Trauma deaths (WHICH DEATHS ARE THESE DROWNINGS, POSIONSINGS, NON INJURY RELATED?)

Avoidable trauma deaths [MOVED]

Nonavoidable trauma deaths [MOVED]

Thank you for the suggestion. We have made these changes to the table and hope that this will also aid in better understanding of the presented data.